# In Silico Comparison Shows that the Pan-Genome of a Dairy-Related Bacterial Culture Collection Covers Most Reactions Annotated to Human Microbiomes

**DOI:** 10.3390/microorganisms8070966

**Published:** 2020-06-27

**Authors:** Thomas Roder, Daniel Wüthrich, Cornelia Bär, Zahra Sattari, Ueli von Ah, Francesca Ronchi, Andrew J. Macpherson, Stephanie C. Ganal-Vonarburg, Rémy Bruggmann, Guy Vergères

**Affiliations:** 1Interfaculty Bioinformatics Unit, University of Bern, Baltzerstrasse 6, CH-3012 Bern, Switzerland; thomas.roder@bioinformatics.unibe.ch (T.R.); danielwue@hotmail.com (D.W.); remy.bruggmann@bioinformatics.unibe.ch (R.B.); 2Swiss Institute of Bioinformatics, University of Bern, CH-3012 Bern, Switzerland; 3Agroscope, Schwarzenburgstrasse 161, CH-3003 Bern, Switzerland; cornelia.baer@agroscope.admin.ch (C.B.); zahra.sattari@dbmr.unibe.ch (Z.S.); ueli.vonah@agroscope.admin.ch (U.v.A.); 4Department for Biomedical Research (DBMR), University Clinic for Visceral Surgery and Medicine, Bern University Hospital, University of Bern, Murtenstrasse 35, CH-3008 Bern, Switzerland; francesca.ronchi@dbmr.unibe.ch (F.R.); andrew.macpherson@dbmr.unibe.ch (A.J.M.); stephanie.ganal@dbmr.unibe.ch (S.C.G.-V.)

**Keywords:** dairy microbiome, human gut microbiome, diversity, health

## Abstract

The diversity of the human microbiome is positively associated with human health. However, this diversity is endangered by Westernized dietary patterns that are characterized by a decreased nutrient variety. Diversity might potentially be improved by promoting dietary patterns rich in microbial strains. Various collections of bacterial cultures resulting from a century of dairy research are readily available worldwide, and could be exploited to contribute towards this end. We have conducted a functional in silico analysis of the metagenome of 24 strains, each representing one of the species in a bacterial culture collection composed of 626 sequenced strains, and compared the pathways potentially covered by this metagenome to the intestinal metagenome of four healthy, although overweight, humans. Remarkably, the pan-genome of the 24 strains covers 89% of the human gut microbiome’s annotated enzymatic reactions. Furthermore, the dairy microbial collection covers biological pathways, such as methylglyoxal degradation, sulfate reduction, γ-aminobutyric (GABA) acid degradation and salicylate degradation, which are differently covered among the four subjects and are involved in a range of cardiometabolic, intestinal, and neurological disorders. We conclude that microbial culture collections derived from dairy research have the genomic potential to complement and restore functional redundancy in human microbiomes.

## 1. Introduction

Modern genomic technologies have deeply impacted the understanding of the functionality of microbial communities in humans [1]. In particular, the gut microbiome modulates the balance between health and disease in a large array of biological phenomena, including the supply of nutrients and energy, intestinal motility, immunity, cardiovascular function, cancer, infections, and many more [2]. These properties are fueled by the diversity of the gut microbiome, which encodes about 150 times more genes [3] than the human genome, including genes for biochemical pathways that are not covered by the human genome, but are relevant for human physiology and maintenance [4].

Biodiversity is a critical aspect of functioning ecosystems, providing them with stability and the ability to respond to external stimuli in a more resilient manner [5]. The same holds true for the human microbiome, as anticipated by microbiologist R. J. Dubos in 1966 [6]. Despite this, the last 50 years witnessed the rise of “Westernized” lifestyles, which are characterized by a decreased diversity in nutrient intake. Concomitantly, many chronic pathologies, such as obesity, type 2 diabetes, and inflammatory bowel disease have become prominent public health issues [7]. A shared feature of these disorders is a reduction in the diversity of the gut microbiome, an observation that has been linked to changes in dietary patterns [8].

People with very different gut microbial composition can be equally healthy, suggesting that the intestinal ecosystem contributes to host homeostasis with a large degree of functional redundancy. However, dysbiosis can appear if functional redundancy is decreased, by the disappearance of taxa that significantly contribute to this homeostasis [9], including so far unknown taxa, which cannot be cultured outside of the intestinal environment. However, functional redundancy could possibly be replenished by direct colonization with the missing taxa, or even taxa other than the originally extinct ones, or by the provision of metabolites promoting the reintroduction of the original species or functional redundant species. As a proof of concept, the extinction of microbial taxa in humanized mice fed with a Western diet over several generations could be reversed by oral administration of the missing taxa [10]. The fact that the oral route taken in this study was successful provides support for human nutritional strategies which aim to deliver bacteria and products of their metabolism to maintain or restore gut homeostasis.

The impact of fermented foods on human health was put forward by Metchnikoff in the early 1900s when he proposed that yoghurt may extend human lifespan [11]. This early work triggered significant scientific efforts during the last century to promote health by using prebiotics and probiotics.

However, the reductionist view of pre-omics science in the 1980s has led food scientists and microbiologists to concentrate their efforts on a narrow range of pre- and probiotic strains rather than on fermented foods with complex microbial composition [12], thus taking the risk to negatively contribute to microbial diversity in human diets. With food being complex, as well as the gut microbiome being primarily characterized by their complexity, it is not a surprise that these narrow strategies were met with moderate success at improving health [13,14].

Meanwhile, the consumption of fermented milk products introduces a diverse range of microbes, which may positively contribute to the restoration of gut microbial diversity [10,15]. Humans have been fermenting milk for almost ten thousand years, primarily to increase its shelf life, but also to be able to tolerate it better and improve its taste. This has likely promoted a close interaction between the ecological niche of lactic acid bacteria (LAB) in dairy environments and the human gut microbiome. Indeed, LAB are acidophilic organisms growing well at pH 3.5–6.5 and constituting about 0.01 to 1.8% of the total bacterial community in the gut. In this respect, recent research indicates that lactic acid bacteria populating the gastrointestinal tract are originating from fermented foods [16]. Studies in mice [17,18] and piglets [19] have revealed that *Lactobacillus* are present in the small intestine. Given the importance of the small intestine for nutrient absorption, targeting functional activities of LAB in this part of the gastrointestinal tract appears to be an interesting nutritional strategy. However, according to the current state of knowledge, only few LAB species seem to stably integrate into the gut microbiome and most require continuous uptake through the diet [20]. Nonetheless, a functional response may be achieved by the high proportion of fermented food and beverage ingested by humans, that is estimated to be between 5% and 40% [15]. Moreover, temporary colonization may be advantageous, by allowing a better control of the health risks associated with the consumption of these organisms.

Apart from potentially being able to complement the microbiome, microbes from fermented foods may also produce or consume compounds with relevance to human health or provide substrates for the resident gut microbiome, leading to metabolites that in turn affect the health of the host. For instance, some *Lactobacillus* strains such as *L. rhamnosus* can digest lactose, which may be advantageous for lactose intolerant individuals [21]. Some LABs can produce folate (vitamin B9), which is a particularly relevant nutrient for pregnant women, as it can help to prevent birth defects such as *spina bifida* [22]. Furthermore, *Lactobacillus* strains such as *L. reuteri* are known for their ability to metabolize tryptophan and produce indole derivatives, which can bind and activate the human aryl hydrocarbon receptor (AhR) [23]. AhR is a transcription factor that has an important role in cellular proliferation and differentiation and adaptive and innate immune response, as well as detoxification [24].

The consortium of bacteria present in dairy culture collections established over decades may thus contain some of the genomic diversity that was lost in modern food production. In this article, we hypothesize that such collections share biochemical functions with the human gut microbiome and might thus represent a strategically interesting source of bacteria to contribute to the establishment of a healthy gut microbiome. As of March 2019, Agroscope, the Swiss center of excellence for agricultural research, had sequenced and annotated 869 strains of its “Liebefeld collection”, containing more than 10,000 isolates, mostly LAB, originating from the Swiss dairy environment. For this analysis, we confine ourselves to one strain per species that might conceivably be used in food production. These 24 strains will hereafter be referred to as the “Liebefeld selection”. To evaluate the potential of the Liebefeld collection to contribute to the functionality of the human gut microbiome, we used in silico methods to compare the coverage of MetaCyc superpathways [25] encoded by the genomes of the Liebefeld selection with the published gut microbiome of four healthy overweight individuals [26]. Given the role of the gut microbiome composition and diversity in obesity [27,28], variations in the genomic content of the gut microbiome of these subjects could provide hints on the replenishing potential of the Liebefeld collection.

## 2. Materials and Methods

### 2.1. Genome Sequencing

The strains were either sequenced using PacBio (19) or Illumina (612) technologies. Library preparation, sequencing and genome assembly were performed as described in Wüthrich et al. [29] In brief, Illumina reads were trimmed with Trimmomatic [30] and assembled with SPAdes [31], whereas PacBio reads were assembled using the HGAP 3 pipeline [32]. Assembly and annotation statistics for the Liebefeld selection strains are reported in Appendix A.

### 2.2. Annotation of the Genome Assemblies

Before sequencing, the strains were taxonomically classified using MALDI-TOF fingerprinting, as described previously [33]. After sequencing, the taxonomic classification was adapted, when appropriate, based on 16S analysis and assembly similarity to genomes available at NCBI [34]. The de novo assemblies were structurally annotated using Prokka (version 1.9) [35]. The predicted coding sequences (CDSs) were blasted against Swiss-Prot [36,37], and the hits (*e*-value < 10^6^) were clustered based on alignment identity and query coverage using the machine learning algorithm DBSCAN [38]. The gene ontology (GO) terms of the cluster containing the best hit were assigned to the CDS. In addition, the GO terms of all found protein families (Pfam) (*e*-value < 10^6^) [39] were assigned to the CDSs. The identified GO terms were then mapped to enzyme commission (EC) numbers. This algorithm, and a more detailed explanation thereof, is available on GitHub [40].

### 2.3. Selection of LAB

From the 869 entries in our sequencing database, we removed legally restricted strains, strain mixtures and strains which could not confidently be categorized taxonomically. Furthermore, assemblies with more than 500 scaffolds and less than 90% BUSCO v3 [41] single-copy-completeness were discarded. Some strains were sequenced multiple times. In this case, duplicates were removed in favor of the better assembly. Each of the 869 entries were shown to describe a unique genome and this report therefore refers to each of them as a “strain” and not an “isolate”, in agreement with van Rossum et al. [42]

From the 31 species, 7 were excluded because they are not relevant to dairy product development: *Brevibacterium linens*, *Corynebacterium variabilis*, *Desemzia incerta* and *Glutamicibacter arilaitensis* occur only in the cheese rind. *Anaerosphaera aminiphila* strains were first isolated from swine manure and are most likely contaminants of raw milk. Furthermore, the genus *Listeria* is associated with disease, and thus, strains that were identified as *Listeria monocytogenes* or *Listeria innocua* were excluded. The 626 remaining strains will hereafter be termed “Liebefeld collection”.

From each of the 24 species remaining in the Liebefeld collection, the strain with the highest number of unique EC numbers (uEC) was selected. These 24 strains will hereafter be referred to by their species name and collectively referred to as “Liebefeld selection”.

### 2.4. Selection of Human Microbiomes

Sequences of four human microbiomes (MH0001-4) were downloaded from GutCyc [26]. The microbiomes belong to middle-aged overweight Danes, two male and two female, living in the northern part of the Copenhagen region. The microbiomes were measured as inflammatory bowel disease (IBD) control subjects for the MetaHIT study. At the original recruitment, the individuals had normal fasting plasma glucose and normal 2-hour plasma glucose, following an oral glucose tolerance test. At the time of fecal sampling, all were examined in the fasting state and had non-diabetic fasting plasma glucose levels below 7.0 mmol/L [3]. The available metadata are summarized in Appendix A. Information about their diet is not available.

Twenty-four genomes were randomly chosen from the 1520 human gut bacteria published by Zou et al. in 2019 [43], and annotated using Prokka (version 1.11) [35]. They were then annotated with EC-numbers, as described in Section 2.2. Their assembly and annotation statistics are reported in Appendix A.

### 2.5. Calculation of Core- and Pan-Genomes

The pan-genome comprises the set of EC-numbers present in any of the respective strains or microbiomes. Similarly, the core-genome comprises the set of EC-numbers that occur in all respective strains or microbiomes.

### 2.6. Calculation of Superpathway Coverage

As a heuristic to assess biochemical potential, we chose MetaCyc over the Kyoto Encyclopedia of Genes and Genomes (KEGG), because MetaCyc has a stronger focus on biochemistry and reactions compared to KEGG, which is more medicine- and compound-oriented. MetaCyc contained more reactions (14,039 EC-annotated reactions) compared to KEGG (11,381 reactions), as of early 2020. MetaCyc superpathways consist of connected sub-pathways providing—compared to KEGG—a more comprehensive biochemical context with regard to scale and purpose [44]. MetaCyc superpathways are annotated with an “expected taxonomic range”, which can be very broad (e.g. “pyrimidine ribonucleosides degradation” in *Archaea*, *Bacteria*, and *Metazoa*), or, on the other hand, consist of rarely studied genes, which are only known to occur in a single species (e.g. “tetrathionate reduction” in *Salmonella typhimurium*). Thus, some superpathways are expected to be covered by most strains, while others may not be covered by any strain or microbiome.

Superpathway coverage was calculated by mapping the annotated EC-numbers (Section 2.2) to the bacterial superpathways of the MetaCyc database (version 22.6) [25], using a python script [45].

Because superpathways consist of multiple pathways, complete superpathway coverage is not always required for the synthesis of biologically important molecules. Since the main functions of the human gut microbiome include energy production from non-digestible carbohydrates, the deconjugation and dehydroxylation of bile acids, the biosynthesis of vitamins and isoprenoids, cholesterol reduction, and the metabolism of amino acids and xenobiotics [4]; much of the functionality of the gut microbiome should be covered by the superpathways.

Furthermore, some superpathway reactions are not annotated with EC-numbers, and can therefore not be completely covered in this analysis, even though the relevant gene may be present.

The superpathway coverage of the “average Liebefeld collection strain” is defined as the average coverage value obtained using every strain from the Liebefeld collection. The superpathway coverage of the “Liebefeld random selection” is defined as the average of 1000 bootstrapped random selections of 24 strains from the Liebefeld collection, without regard to taxonomic classification.

## 3. Results and Discussion

### 3.1. Liebefeld Collection Overview

Figure 1 presents, for each strain, the number of genes, as well as the EC annotation rate. It provides information on the scale (number of strains), taxonomic composition (number of species), and diversity (differing numbers of genes and annotation rates between species) of the Liebefeld collection.

The Liebefeld collection strains have fewer genes than the 24 randomly selected gut microbes (Appendix A). The gut microbes have, on average, 3497 genes, which is slightly more than the Liebefeld collection strain, with the highest number of genes. Further, the distributions of the annotation rate are significantly different (Mann–Whitney U test, p-value = 1.68 × 10^−4^), the Liebefeld selection demonstrating higher annotation rates (Appendix A).

Unsurprisingly, the number of genes correlates strongly with genome size (R^2^ = 90%) and, consequently, with the species. While genomes from the same species have similar numbers of genes, there is a large range within the genus *Lactobacillus*. The number of genes correlates with the number of EC-annotated genes, but the EC annotation rate also depends on the species. For example, *Lactobacillus helveticus* strains, despite having a similar number of genes as *Propionibacterium freudenreichii* strains, have a significantly lower annotation rate.

### 3.2. Comparison of Superpathway Coverage

As a display of the biochemical potential and functional diversity of the Liebefeld collection, Figure 2 shows the superpathway coverage of the individual strains of the Liebefeld selection in comparison with the four human microbiomes. The average superpathway coverage of the Liebefeld selection strains ranged from 36% (*Lactobacillus helveticus*) to 53% (*Acidipropionibacterium acidipropionici*). The average Liebefeld collection strain covered 42% of the superpathways. Remarkably, the pan-genome of the Liebefeld selection covered 65% of the superpathways, and the human microbiomes between 60% (MH0004) and 67% (MH0003), indicating that 24 strains suffice to reach a coverage similar to the human microbiome.

Furthermore, the average superpathway coverage of the 24 Liebefeld selection strains (Liebefeld selection: pan genome) was significantly higher than that of 24 randomly selected strains (Liebefeld random selection: pan genome) (*p*-value under normal distribution = 1.58 × 10^−4^).

In all strains and microbiomes, the majority of superpathways remained only partially covered. No single strain covered more than 27 superpathways completely. Together, the Liebefeld selection covered 39 superpathways completely, placing it within the range of human microbiomes (34 to 44).

The 24 randomly selected genomes of human gut bacteria have a slightly higher mean superpathway coverage than the Liebefeld selection strains (Appendix A), but the difference in their distribution is not significant (Mann–Whitney U test, *p*-value = 0.32).

The detailed coverage of each superpathway, as well as dendrograms, is available in Figure 3.

### 3.3. Comparison of Unique EC Numbers (uECs)

Figure 4 gives an overview of the shared genomic content between the strains of the Liebefeld selection and the four human microbiomes. The number of unique EC-numbers (uECs) that were annotated to the strains of the Liebefeld selection range from 727 (*Lactobacillus helveticus*) to 1161 (*Acidipropionibacterium acidipropionici*). Since it would be beyond the scope of this study to go into detail about the enzymes and reactions covered, this analysis is limited to a numerical comparison. On average, a single strain of the Liebefeld selection covers 53% of the uECs of the human microbiomes (MH0001: 59%; MH0002: 49%; MH0003: 49%; MH0004: 56%). Taken together, the 24 strains of the Liebefeld selection contain 1676 uECs, a number comparable to that of the studied human microbiomes, their uECs ranging from 1367 (MH0001) to 1811 (MH0003). On average, the Liebefeld selection pan-genome covers 89% of the uECs of the four studied human microbiomes (MH0001: 92%; MH0002: 87%; MH0003: 87%; MH0004: 90%). Taken together, the 626 strains of the Liebefeld collection contain 1728 uECs, i.e., more than two of the four studied microbiomes, and they cover 91% of the uECs of the four human microbiomes (MH0001: 94%; MH0002: 89%; MH0003: 89%; MH0004: 92%). Conversely, if the 24 strains of the Liebefeld selection were added to the four human microbiomes, the latter would gain, on average, 17% uECs (MH0001: 31%; MH0002: 8%; MH0003: 6%; MH0004: 24%). However, because of the variability of the four human microbiomes, only few uECs in the Liebefeld collection (5%) and the Liebefeld selection (4%) cannot be found in any of the four studied human microbiomes. Figure 5 graphically illustrates the large overlap of uECs between the human microbiome and the Liebefeld selection, as well as the Liebefeld collection.

In addition, the 24 Liebefeld selection strains (Liebefeld selection: pan genome) have significantly more uECs than 24 randomly selected strains (Liebefeld random selection: pan genome) (*p*-value under normal distribution = 2.53 × 10^−6^). Even though we assume that organisms have a large number of similar enzymes in common, the biochemical potential of the Liebefeld collection is remarkable. The number of uECs present in its pan-genome exceeds that of two out of the four studied human microbiomes and resulted in a MetaCyc superpathway coverage greater than three of them. Even after restricting the number of strains to 24, each from a different species (Liebefeld selection), the pan-genome showed a higher superpathway coverage and more uECs than any of the 1’000 randomly chosen combinations of 24 strains (Liebefeld random selection). Furthermore, their superpathway coverage, as well as the number of uECs, was well within the range of the human microbiomes analyzed in this study. Thus, the results of our in silico study on the biochemical potential of the strains that originate from the dairy environment are promising. However, these results could be improved, as only one strain per species was selected to build the Liebefeld selection.

### 3.4. Functional Properties of the Gut Microbiome, Which Might be Enriched by the Liebefeld Collection

The four human subjects selected in this report were healthy, although overweight. Individual differences in the coverage of their MetaCyc superpathways may be associated with their metabolic or health status, possibly indicating the onset of dysbiosis. To illustrate our approach, we have searched for such superpathways in the microbiomes of the four subjects and addressed whether the underrepresented microbial functions might theoretically be supported by the Liebefeld selection or strains thereof. The fifteen superpathways with the highest variance amongst the human microbiomes are presented in Figure 6, and some of them are discussed in their biological context below.

#### 3.4.1. Methylglyoxal Degradation IV Superpathway

The superpathway “methylglyoxal degradation IV” has a low coverage in subjects MH0001 and MH0004 (Figure 6). Methylglyoxal is a product of glucose and glycine metabolism that increases the activity of the gut microbial trimethylamine (TMA)-lyase [46]. TMA lyase catalyzes the transformation of dietary choline and carnitine to TMA, which in turn is metabolized by the liver to trimethylamine-N-oxide (TMAO), a potential risk factor for cardiovascular diseases [47]. The pan-genome of the Liebefeld selection covers the superpathway “methylglyoxal degradation IV” similarly to subjects MH0002 and MH0003. Among the 24 species of the selection, nine strains demonstrate a high coverage of the superpathway (L*. parafarraginis, L. parabuchneri, L. delbrueckii ssp. lactis, L. delbrueckii ssp. bulgaricus, Facklamia tabacinasalis, L. paracasei, L. casei, Staphylococcus xylosus, and Acidopropionibacterium acidopropionici*). These strains could enhance the methylglyoxal degradation capability in the subjects MH0001 and MH0004, and potentially lower their TMAO levels. That the fermentation of food products with LAB can redirect the transformation of precursors of TMAO was recently demonstrated by Burton et al. [48], who showed that the fermentation of milk to yoghurt decreases TMAO in urine and plasma.

#### 3.4.2. Assimilatory Sulfate Reduction I Superpathway

The superpathway “Assimilatory sulfate reduction I” has a lower coverage in subjects MH0001 and MH0004 (Figure 6). Intestinal microorganisms use sulfate to synthesize cysteine via the assimilatory sulfate reduction pathway [49,50]. Sulfate can, however, also be metabolized via the dissimilatory sulfate reduction pathway to produce hydrogen sulfide (H_2_S). H_2_S is a toxic molecule associated, among others, with IBD. On the other hand, recent research has revealed that, similar to nitric oxide (NO), H_2_S is an important signaling molecule, with therapeutic potential in a range of diseases, in particular oxidative stress-induced neurodegenerative diseases [51]. The pan-genome of the Liebefeld selection covers the superpathway “Assimilatory sulfate reduction I”, similarly to subjects MH0002 and MH0003, and could thus potentially shift the activity of the gut microbiome towards assimilatory sulfate reduction. As H_2_S inhibits the growth of LAB strains, hampering their development as probiotics [52], LAB strains diverting sulfate metabolism towards the assimilatory pathway could be interesting components of probiotic products. Among the 24 species of the selection, one strain demonstrates a high coverage of this superpathway (*Lactobacillus plantarum*). 

#### 3.4.3. 4-Aminobutanoate Degradation (GABA) Degradation Superpathway

The superpathway “4-aminobutanoate degradation” has a lower coverage in subject MH0004 compared to the other three subjects (Figure 6). The inhibitory neurotransmitter 4-aminobutanoate (GABA), which is also synthesized by microbes in the intestine, is known for balancing the stimulation of synapses by glutamate in the brain. Although, to our knowledge, there are no studies which show direct evidence for the effect of gut-derived GABA on the human CNS, an in vivo study in mice with GABA-producing *Lactobacillus rhamnosus (JB-1)* revealed changes in the mRNA of GABA receptors B1b and A2, as well as reduced anxiety- and depression-related behavior [53]. Specific GABA/glutamate antiporters mainly achieve homeostasis between glutamate and GABA, although the degradation of GABA also plays a role. The pan-genome of the Liebefeld selection covers the superpathway “GABA degradation”, similarly to the microbiome of subjects MH0001, MH0002, and MH0003. Among the 24 species of the selection, five strains demonstrate a high coverage of this superpathway (*Staphylococcus xylosus, L. parafarraginis, Propionibacterium freudenreichii, Acidopropionibacterium acidopropionici, and Acidopropionibacterium jensenii*).

#### 3.4.4. Salicylate Degradation Superpathway

The superpathway “salicylate degradation” has a lower coverage in subject MH0001 and MH0004 compared to the other two subjects (Figure 6). Salicylates are known for their analgesic, antipyretic, antithrombotic and anti-inflammatory effects. The main mechanism by which these effects are achieved is the inhibition of cyclooxygenases, which are responsible for the biosynthesis of prostaglandins. However, as recent research shows, salicylates also induce the activation of the adenosine monophosphate-activated protein kinase (AMPK) [54], which plays an important role in cellular energy homeostasis and immunity, promoting the generation of Tregs [55]. However, salicylates can also activate the AhR [56], which is involved in cellular proliferation and differentiation and has a major role in adaptive and innate immune response [57]. Furthermore, salicylates influence the intestinal microbiome itself by decreasing the expression of adherence factors and biofilm formation [58]. The degradation of salicylates therefore contributes to the salicylate homeostasis and thus influences different biological functions regulated by AMPK and AhR activation, as well as the composition and structure of the intestinal microbiome. The pan-genome of the Liebefeld selection covers the superpathway “salicylate degradation” similarly to subjects MH0002 and MH0003. Among the 24 species of the selection, only one strain, *Staphylococcus xylosus,* shows a higher coverage as MH0001 and MH0004, but seven further strains demonstrate a comparable high coverage of the superpathway (L. *fermentum, Leuconostoc mesenteroides, Lactococcus lactis ssp. lactis, L. parafarraginis, L. paracasei, L. casei, L. rhamnosus*).

### 3.5. Limitations

The usefulness of the bioinformatic analysis in this study depends on the quality of the annotations. Because ubiquitous genes are better studied and annotated than less common genes, the extent of overlap between the Liebefeld strains and the microbiomes is likely overestimated. In addition, the human microbiome has received little attention from researchers until recent years. Since it consists of mostly non-cultivable species, the available information on the composition and dynamics of the species present is still limited. As the genomes of well-studied species are likely better annotated (see Appendix A for a comparison of annotation rate between the Liebefeld selection and the 24 randomly selected gut bacteria), it is possible that the superpathway coverage of the four human microbiomes is underestimated.

To achieve a sustainable effect on the human microbiome, a strain must either be able to integrate stably into the microbiome or be supplied continuously. For stable integration, factors that favor inclusion into the gut microbiome must be considered, such as resistance to low pH, ability to tolerate bile salts, pancreatin, pepsin, lysozyme, and H_2_S [52], the ability to compete with other gut bacteria, and the ability to adhere to intestinal epithelial cells or mucus [59,60]. It may be possible to predict some of these factors by studying the genomes of the strains. For example, genes that belong to the mucus-binding (MUB) protein family, which has first been discovered in *Lactobacillus reuteri* and *acidophilus*, could be indicators of adaptation to the gut environment. Pili have also been shown to mediate the gut adhesion of *Bifidobacteria*, but these structures have a far wider applicability and are probably less sensitive predictors [60]. In this context, although most species in the Liebefeld selection are not classified as common residents, most have been detected in the human gastrointestinal microbiome (Appendix A) [61,62,63].

However, the presence of such genes does not necessarily translate into their expression in the relevant environment, and as most relevant genes are unknown in the first place, biological experiments are indispensable. This study focuses on the presence or absence of annotations without regard to gene copy numbers and similar nuances. As gene duplication often increases gene dosage [64] and contributes to the diversification of microbes in the gut [65], it could be interesting to add this parameter to the panel of criteria to identify interesting bacteria. This analysis goes, however, beyond the scope of this report.

Although Section 3.4 provides an illustration of the functional potential of the Liebefeld collection, these examples are limited to healthy overweight individuals. This approach could be extended to a comparison of the microbiomes of healthy and dysbiotic patients. Such an analysis would, however, also require a large dataset, in order to move from the illustrative cases presented in Section 3.4 to a clinically meaningful approach. In addition, the complementation with RNA-seq data would also be desirable to measure whether, and in what proportion, the genes of interest are actually expressed.

## 4. Conclusions

Our in silico study shows that the Liebefeld collection, which consists of strains of LAB that were collected in the Swiss dairy environment during a period of almost a century, has significant biochemical potential. In particular, the Liebefeld collection offers a strategic opportunity to design food products, which could possibly mitigate the negative effects of Westernized diets on health, by restoring the functional redundancy of the gut microbiome, which is often lost in the context of chronic diseases like obesity, diabetes, and inflammatory bowel disease. Currently, the potential of these strains is explored for various applications in food production. In this context, bioinformatic investigations are a powerful tool for identifying the most promising candidates at the genetic level. In particular, the selection of combinations of several strains that complement each other at the biochemical level is a novel and highly interesting concept for the food industry. The approach shown in this study thus has the potential to revolutionize the production in the field of fermented foods and lead to a completely new product palette that meets the growing demand for health supporting foods.

## Figures and Tables

**Figure 1 microorganisms-08-00966-f001:**
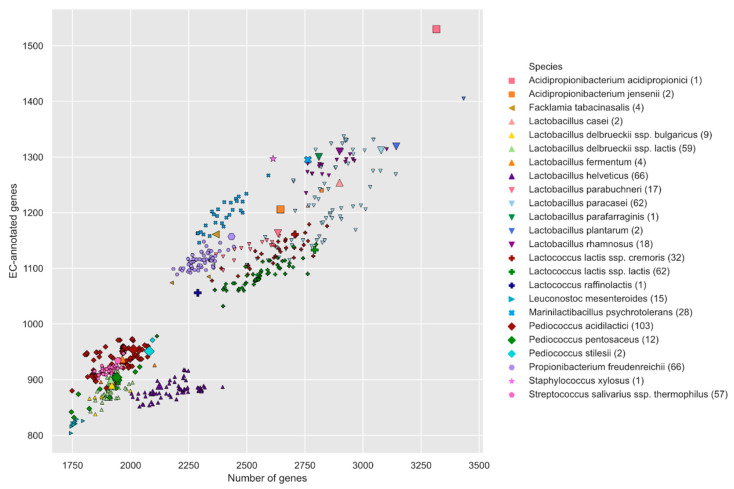
Relationship between the number of identified genes and the number of enzyme commission (EC)-annotated genes per Liebefeld collection strain. Strains from the Liebefeld selection are highlighted with a larger symbol. Each species is represented by a unique symbol and color. A plot that includes the 24 human gut bacteria randomly selected from Zou et al. [43] is available in Appendix A.

**Figure 2 microorganisms-08-00966-f002:**
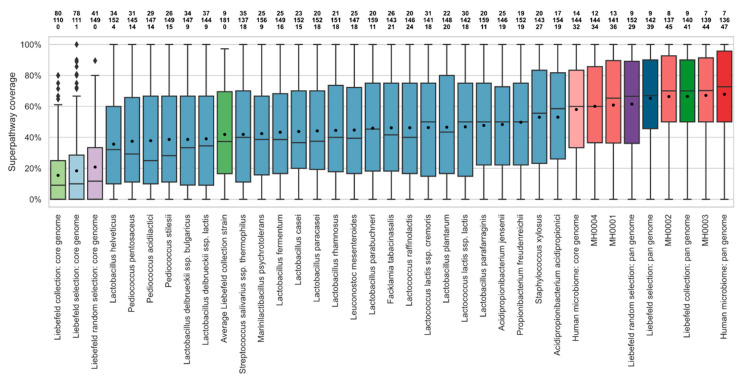
Boxplot of the coverage of the 190 MetaCyc superpathways by each of the 24 strains of the Liebefeld selection (blue, referred to by their species name) and the four human microbiomes (red, MH0001-4), the Liebefeld collection (green), and the Liebefeld random selection (violet). Core-genomes are colored in a lighter and pan-genomes in a darker shade of the corresponding color. The strains or sets of strains are sorted in ascending order according to their mean superpathway coverage, indicated by a black dot. Above each boxplot, three numbers indicate how many superpathways are not covered (top row), partially covered (middle row) and completely covered (bottom row). An analogous plot comparing the Liebefeld selection strains to the 24 human gut bacteria randomly selected from Zou et al. [43] is available in Appendix A.

**Figure 3 microorganisms-08-00966-f003:**
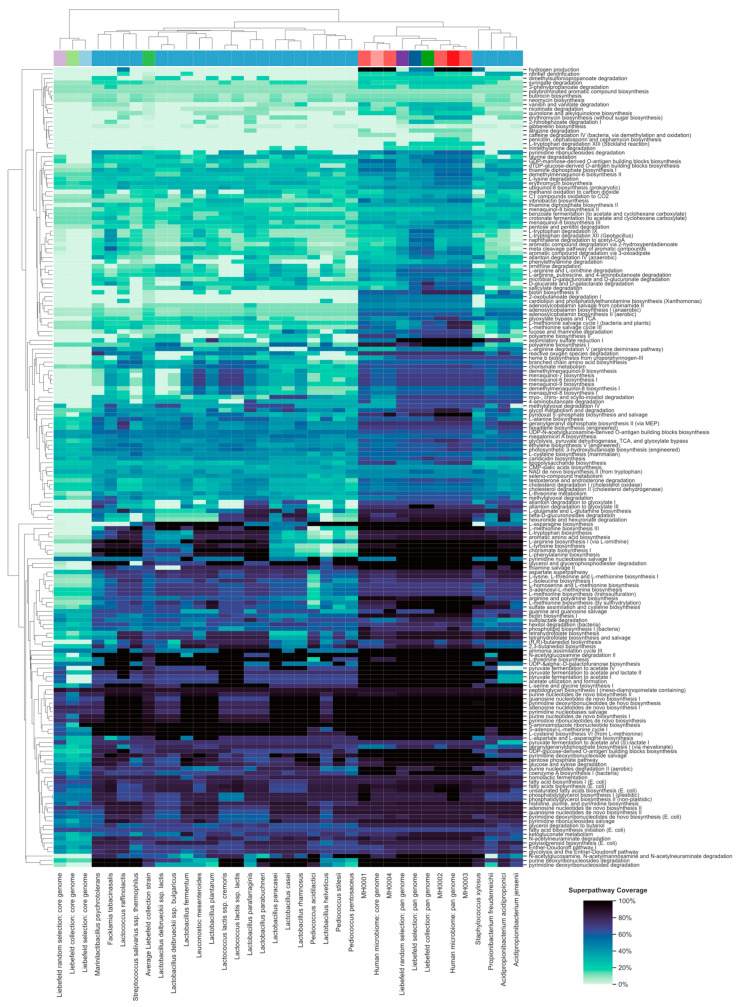
Overview of the biochemical potential of the 24 strains of Liebefeld selection (blue, referred to by their species name), the four human microbiomes (red, MH0001-4), the Liebefeld collection (green), and the Liebefeld random selection (violet). Core-genomes are colored in a lighter- and pan-genomes in a darker shade of the corresponding color. The Y-axis denotes the 190 superpathways of MetaCyc. The dendrogram of both axes resulted from hierarchical clustering. The colors of the heatmap denote superpathway coverage and range from white (0%) to black (100%). An analogous plot comparing the Liebefeld selection strains to the 24 human gut bacteria randomly selected from Zou et al. [43] is available in Appendix A.

**Figure 4 microorganisms-08-00966-f004:**
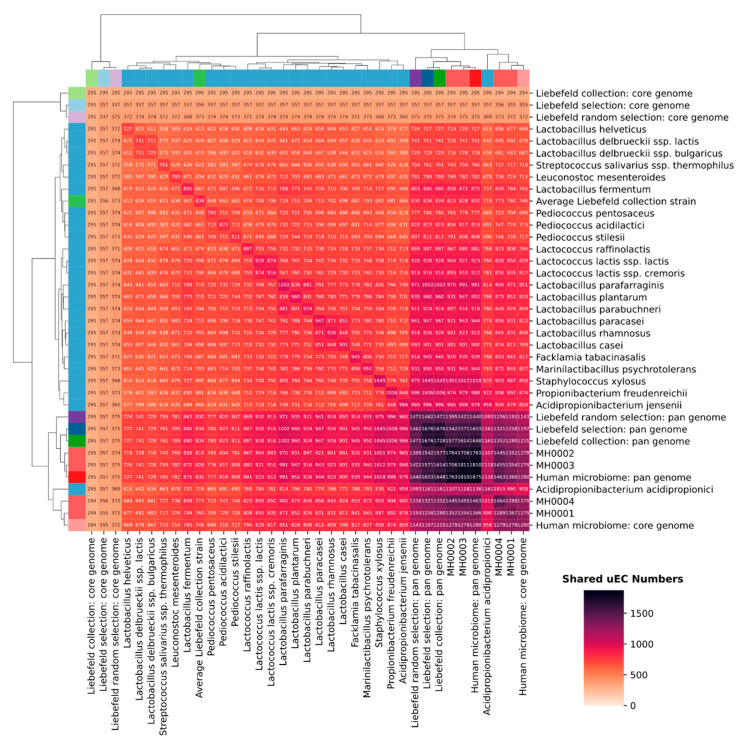
Clustered heatmap of the number of shared unique EC numbers (uECs) of 24 strains of Liebefeld selection (blue, referred to by their species name), the four human microbiomes (red, MH0001-4), the Liebefeld collection (green), and the Liebefeld random selection (violet). Core-genomes are colored in a lighter- and pan-genomes in a darker shade of the corresponding color. The total number of uECs annotated to a species/metagenome can be read from the antidiagonal line, where the same species/metagenome intersect. The dendrogram of both axes resulted from hierarchical clustering. An analogous plot comparing the Liebefeld selection strains to the 24 human gut bacteria randomly selected from Zou et al. [43] is available in Appendix A.

**Figure 5 microorganisms-08-00966-f005:**
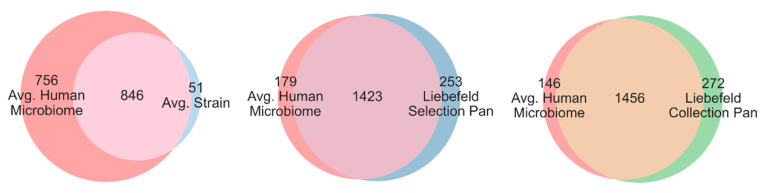
Venn diagrams of the shared unique EC numbers (uECs) between the average human microbiome (red) and the average Liebefeld selection strain (light blue), the Liebefeld selection pan-genome (dark blue) and the Liebefeld collection pan-genome (green).

**Figure 6 microorganisms-08-00966-f006:**
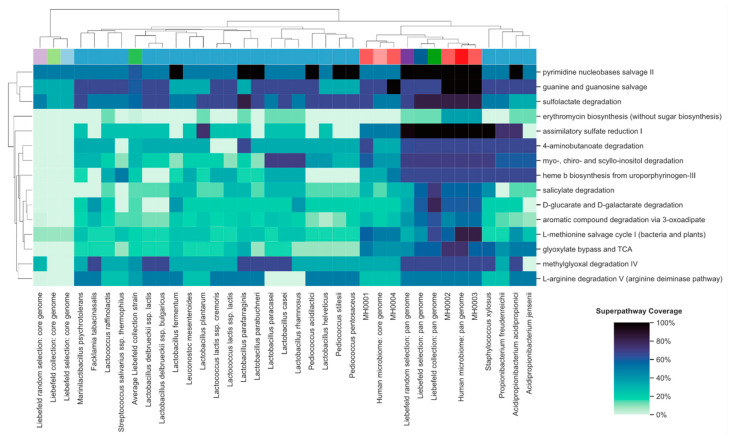
Overview of the biochemical potential of the 24 strains of Liebefeld selection (blue, referred to by their species name) and the four human microbiomes (red, MH0001-4), the Liebefeld collection (green), and the Liebefeld random selection (violet), for the fifteen superpathways with the highest variance amongst the human microbiomes. Core-genomes are colored in a lighter and pan-genomes in a darker shade of the corresponding color. The Y-axis denotes these 15 superpathways. The dendrogram of both axes resulted from hierarchical clustering. The colors of the heatmap denote superpathway coverage and range from white (0%) to black (100%). An analogous plot comparing the Liebefeld selection strains to the 24 human gut bacteria randomly selected from Zou et al. [43] is available in Appendix A.

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
