# Peer review of "In Silico Comparison Shows that the Pan-Genome of a Dairy-Related Bacterial Culture Collection Covers Most Reactions Annotated to Human Microbiomes"

_microorganisms, 2020, doi:10.3390/microorganisms8070966_

Round 1
Reviewer 1 Report
The authors addressed most of my concerns.
Some pending comments
3rd paragraph of introduction (line 64-75) could be improved.
Figure text size can be improved.
Author Response
The authors would like to thank reviewer 1 for the very useful comments on our revised manuscript, which helped us to further improve the manuscript. Responses to the detailed remarks are given below.
3rd paragraph of introduction (line 64-75) could be improved.
We have clarified the text as follows (lines 65-79)
However, the reductionist view of pre-omics science in the 1980s has led food scientists and microbiologists to concentrate their efforts on a narrow range of pre- and probiotic strains rather than on fermented foods with complex microbial composition [12], thus taking the risk to negatively contribute to microbial diversity in human diets. The composition of foods as well as the gut microbiome being primarily characterized by their complexity, it is therefore not a surprise that these narrow strategies were met with moderate success to improve health [13,14].
On the contrary, consumption of fermented milk products introduces a diverse range of microbes, which may positively contribute to the restoration of gut microbial diversity [10,15]. Humans have been fermenting milk for almost ten thousand years, primarily to increase its shelf life, but also to be able to tolerate better and improve its taste. This has likely promoted a close interaction between the ecological niche of lactic acid bacteria (LAB) in dairy environments and the human gut microbiome. Indeed, LAB are acidophilic organisms growing well at pH 3.5–6.5 and constituting about 0.01 to 1.8% of the total bacterial community in the gut. In this respect, recent research indicates that lactic acid bacteria populating the gastrointestinal tract are originating from fermented foods [16].
Figure text size can be improved.
We agree that the text size of the captions and regular text are too similar. However, they correspond to the formatting guidelines of the journal. Therefore, we will discuss this issue with the technical editor if necessary.
Reviewer 2 Report
General Comments:
Roder et al. present a revised manuscript that significantly improves the description of methodology and sample source for comparisons of the biochemical potential of the Liebfeld collection/selection to a set of four publicly -available metagenomes. The authors have now added significant discussion of the limitations of their approach, and the possibility of a consortium of Liebfeld organisms to replete specific biochemical functions in the human metagenomes analyzed. The manuscript seems now in two parts: one where the collection is described and one where applications are discussed. That said, major issues mentioned in my previous review remain with the first part of the paper, plus, the application section of the paper, although an appropriate consideration, is far too speculative and based on a comparison of healthy individuals to each other (rather than healthy to diseased/dysbiotic) and thus significantly weakens the paper.
Specific Comments (limited):
1) The authors have clarified their definition of "strains" in their response. However, I maintain that the use of strains through the manuscript will mislead many readers as opposed to the verbiage of "isolates", "taxa/taxon", or "organisms." Perhaps this distinction is not as significant to a non-microbiologist reader, but this is a general issue with modern taxonomy that must either be clearly defined or avoided.
2) Section 2.2 - Description of Assembly Annotation - is lacking significant details regarding definition of "assembly similarity" and thresholds for search hit acceptance. As written, "best hit" selection for annotation can be VERY misleading.
3) Figure 1 remains a non-novel result. It is intuitive that a reader would appreciate the relationship between gene count and EC annotation and the variance of this relationship between different taxa. Figure 2 is much more in line with the stated title of the manuscript.
4) I appreciate the addition of figures discussing the EC-based similarity of Liebfeld collection/selection pan-genome/core-genome to the reference (meta)genomes. This is again not a terribly novel result, as it's expected that organisms will share large numbers of similar enzymes. A percent of shared content is really a rudimentary analysis without a discussion of what those enzymes are or what is NOT shared (although I realize that is not the point of the paper).
5) Discussion of "missing" functions that may be repleted by Liebfeld organism supplementation. I firmly believe this analysis is based on flawed reasoning, as it identifies biochemical targets based on a comparison of healthy (though overweight) individuals. There is significant literature and methodology considering dysbiotic states associated with disease. The way this analysis is conducted would actually argue that the "missing" functions discussed in detail here are superfluous to gut health.
Author Response
The authors would like to thank reviewer 2 for the thorough review of the revised manuscript and the valuable comments, which helped us to further improve the manuscript. Please find the responses to the specific comments below.
1) The authors have clarified their definition of "strains" in their response. However, I maintain that the use of strains through the manuscript will mislead many readers as opposed to the verbiage of "isolates", "taxa/taxon", or "organisms." Perhaps this distinction is not as significant to a non-microbiologist reader, but this is a general issue with modern taxonomy that must either be clearly defined or avoided.
The following sentence was added to explain the use of the wording “strain” in our report (lines 138-140): “Each of the 869 entries were shown to describe a unique genome and this report therefore refers to each of them as “strain” and not “isolate” in agreement with van Rossum et al. [42].”
2) Section 2.2 - Description of Assembly Annotation - is lacking significant details regarding definition of "assembly similarity" and thresholds for search hit acceptance. As written, "best hit" selection for annotation can be VERY misleading.
Section 2.2. Annotation of the genome assemblies was updated and now contains the information about the BLAST threshold (line 127). For further details, we refer to the GitHub page, where the algorithm is described in more detail and a paper draft can be found (lines 131-132). However, since the Liebefeld collection has been curated with different tools and criteria over the years, we cannot provide a concise description of the "assembly similarity".
3) Figure 1 remains a non-novel result. It is intuitive that a reader would appreciate the relationship between gene count and EC annotation and the variance of this relationship between different taxa. Figure 2 is much more in line with the stated title of the manuscript.
We agree with this comment and changed the introduction to Figure 1 (lines 195-198) to better explain its purpose and reordered the paragraphs below it (lines 204-214) according to their relevance.
4) I appreciate the addition of figures discussing the EC-based similarity of Liebfeld collection/selection pan-genome/core-genome to the reference (meta)genomes. This is again not a terribly novel result, as it's expected that organisms will share large numbers of similar enzymes. A percent of shared content is really a rudimentary analysis without a discussion of what those enzymes are or what is NOT shared (although I realize that is not the point of the paper).
We now explain in section 3.3 that an analysis of individual enzymes and covered reactions is beyond the scope of this publication and that we have therefore limited ourselves to a numerical comparison (lines 258-259). Furthermore, to set our results in the right context, we now mention in section 3.3 that we assume that organisms have a large number of similar enzymes in common (lines 290-291).
5) Discussion of "missing" functions that may be repleted by Liebfeld organism supplementation. I firmly believe this analysis is based on flawed reasoning, as it identifies biochemical targets based on a comparison of healthy (though overweight) individuals. There is significant literature and methodology considering dysbiotic states associated with disease. The way this analysis is conducted would actually argue that the "missing" functions discussed in detail here are superfluous to gut health.
To avoid misinterpretation, we now specify both in sections 3.4 (line 304) and 3.5 (line 411) that this first analysis is only intended to illustrate our strategy. However, since the four humans were overweight, a lack of superpathway coverage could indicate the onset of dysbiosis. We have added this comment in section 3.4 (line 304).
In the last revision, we changed our argument from replacing missing functions to restoring functional redundancy. However, the reviewer is correct in pointing out that some vestiges of the old reasoning remain. Consequently, we updated the title of section 3.4 (line 301) and made the following changes:
- “underrepresented” instead of “missing” (line 306)
- “supported” instead of “replaced (line 306)
- “biological context” instead of “clinical context” (line 308)
- “enhance” instead of “compensate” (line 330).
Moreover, in section 3.5 Limitations, we added a paragraph to outline possible next steps towards a comprehensive analysis with potential clinical significance (lines 411-417).
Round 2
Reviewer 2 Report
I appreciate the authors efforts to explain methodology and reasoning, along with justification for analyses which were not conducted. I do believe that this collection of microorganisms/genomic information is a valuable resource to the research community and should be published.
This manuscript is a resubmission of an earlier submission. The following is a list of the peer review reports and author responses from that submission.
Round 1
Reviewer 1 Report
Roder et al, present a manuscript comparing the functional capacity of dairy bacteria to human gut commensal. They sequenced over 24 dairy strains from the Liebefeld Collection using whole shotgun metagenomics and compared them to 4 human publicly available pan genome.
They report that the coverage and biochemical potential of the Liebefeld Collection pan genome is comparable to the one observed in human samples. According to these results, the authors claim that dairy bacteria could be used to re-colonise and re-diversify the gut microbiota.
However, to make such claim a few points would need to be further developed:
Looking at coverage is interesting; however, number of gene copies would also be very informative in this context.
Although the dairy bacteria may present similar biochemical potential than gut commensal, their use to re-populate the human gut may be limited by numerous factors including, their ability to adapt/proliferate/engraft in the gut, the potential doesn't imply that the gene is functional or used in the necessary context by the bacteria. These limitations should be extensively discussed in the conclusion.
Reviewer 2 Report
The authors investigated whether the metagenome of a set of 24 species of bacteria found in Swiss dairy environment can recapitulate the metabolic capabilities of a healthy gut microbiome. Although the hypothesis addressed in this manuscript is interesting, many aspects of this paper demands major revision to be suitable for publication. One key missing aspect is what is the null expectation of metabolic coverage from random genomes (e.g. in terms of EC number) and a comparison between health and disease states.
The title could be improved. Introduction should bring some information of expected metabolic pathways and capability among human samples What would be the match of random genomes in term of superpathways, biochemical potential and shared EC numbers? The author should educate the readers about the meaning of different pathways. Are some pathways essential in most microorganisms? Are some pathways with direct link with human host metabolism, health or disease? What is the prevalence of the selected 24 strains in human metagenome? What is the impact of metabolic capabilities in health or disease? Can the authors compare the metabolic capabilities (superpathways, biochemical potential and shared EC numbers) among patients from health and disease state (e.g. http://www.gutcyc.org/, Hahn Aria S. et al. - reference 13 in your manuscript)? How do metagenomics capabilities compare among population of different diet type (e.g. western and non-western groups). The authors should contextualize more the impact and relevance of their research in the discussion
Reviewer 3 Report
Summary:
Roder et al describe an investigation into the biochemical capacity and diversity of a collection and subset thereof of a set of genome-sequenced dairy microorganisms (primarily LABs). This collection is presented as a potentially valuable source of isolates for supplementation as possible probiotics. The authors describe the genome sequencing, functional annotation, and exploration of the biochemical capacity of the isolates, referenced to Swiss-Prot, Pfam, and MetaCyc and compared in parallel with a set of sequenced human microbiomes. In all, they describe a rich set of bacterial genes in their collection that capture similar functional repertoire (MetaCyc superpathways and ECs) and diversity as the human microbiomes.
General Comments:
In all, I believe this work has scientific merit and the approach is generally sound. There are key gaps in the explanation of methods and the purpose of specific figures that require modification. Further, the microbiomes presented as a reference to the Liebfeld collection are completely undescribed and as are not a relevant control without further detail.
1) Throughout the manuscript, the authors equate the supplementation of organisms from the Liebfeld collection to restoration of "missing" microbial taxa from the human microbiota. However, the types of organisms common to dairy products are not often common residents of the healthy, adult gut microbiota, nor do probiotic strains (particularly LABs) tend to colonize and persist well in gut. I feel the authors need to address this key conceptual link in their introduction or adjust their introduction to discuss the probiotic concept more directly.
2) The human microbiomes are completed undescribed. Whether they were sequenced by the authors or obtained from a public database, their origin MUST be explained if they are to be used as reference. Further, they are described as "healthy," but details on the age, dietary behaviors, etc. of the donors is essential to their inclusion as a worthy reference.
3) The summarization of quality control and the reasoning for the analyses depicted in Figs 1 and 2 is unclear. We expect a relationship between number of genes and number of ECs. These Figures are better suited to conversion to a summary table or supplementary Figures.
Specific comments:
1) Line 3 - "covers the genomic content" is ambiguous. Consider alternate wording.
2) Line 18 - "dietary patterns rich in microbial species" requires clarification. Do you mean probiotic supplements or are you equating dietary diversity with microbial diversity?
3) Line 27- EC coverage does not equate to pathway functionality without an indication of completeness.
4) Line 53 - "80s" is extremely colloquial (I know this is editorial).
5) Line 63-65 - see general point 1 above.
6) Line 79-83 - Please provide a clearer description of sequencing/assembly strategies. What assembler was used? How much data was obtained and using what library preparation protocol. Were any of the isolates sequences with BOTH PacBio and Illumina?
7) Line 86 - please define CDS at first usage.
8) Line 87 - How was the cutoff for top hits determined? How diverse were these top 20 GO annotations? Can the authors justify the selection of a hard cutoff versus, selecting the "best" hits, subject to ties, for annotation?
9) Line 92 - Will the python script be available on publication?
10) Line 94 - Please describe taxonomic assignment process.
11) Line 104 - How were "strains" defined? Do the authors mean isolates, or did they compare whole-genome sequences?
12) Line 106-109 - Define "group." Taxonomic group? Based on name or phylogenetic relatedness?
13) Figure 1 - The plot is very difficult to parse. Perhaps a larger point diameter would allow the reader to see relationships between isolate genomes in the Liebfeld collection/selection and taxonomic assignments. Is there a relationship between taxonomy and EC annotation success?
14) Figure 2 - I don't think this information warrants an independent Figure. It should be combined with Fig 1 or presented as a supp table.
15) Line 126 - The use and strategy of superpathways as a functional reference should be defined. Does a superpathway have to be completely covered by gene annotations to be "counted?" How many superpathways were completely captured?
16) Figure 4 - Please defined superpathway coverage. Given the size of this heatmap, a reader would benefit from additional annotation. I would strongly suggest not a higher order organization of the y-axis components (as does KEGG in function/pathway/category, etc.) and the removal of any superfamilies that are not at all covered or with very low coverage.
17) Lines 146-162 - the discussion of uECs and comparison to the diversity in the human microbiomes (see general point 2 above) is unclear without more detail in the human microbiomes. Further, are there particular EC types or families that are missing in this specific group of LABs? I believe the authors hint at this in Lines 160-162, specifically.
18) Lines 170-174. I agree that the diversity/count of ECs is high. The key point that they reflect the identity of those in the human microbiomes is much more powerful.
19) Lines 186-190 - Author AM is missing from the contributions list.
